# Explaining and Mitigating the Modality Gap in Contrastive Multimodal Learning

Can Yaras,* Siyi Chen*, Peng Wang, Qing Qu
University of Michigan
{cjyaras, siyich, pengwa, qingqu}@umich.edu

Multimodal learning has recently gained significant popularity, demonstrating impressive performance across various zero-shot classification tasks and a range of perceptive and generative applications. Models such as Contrastive Language–Image Pretraining (CLIP) are designed to bridge different modalities, such as images and text, by learning a shared representation space through contrastive learning. Despite their success, the working mechanisms of multimodal learning remain poorly understood. Notably, these models often exhibit a *modality gap*, where different modalities occupy distinct regions within the shared representation space. In this work, we conduct an in-depth analysis of the emergence of modality gap by characterizing the gradient flow learning dynamics. Specifically, we identify the critical roles of mismatched data pairs and a learnable temperature parameter in causing and perpetuating the modality gap during training. Furthermore, our theoretical insights are validated through experiments on practical CLIP models. These findings provide principled guidance for mitigating the modality gap, including strategies such as appropriate temperature scheduling and modality swapping. Additionally, we demonstrate that closing the modality gap leads to improved performance on tasks such as image-text retrieval.

## 1. Introduction

Recently, significant progress has been made in multimodal learning, particularly in connecting text and image modalities through self-supervised methods that leverage large-scale paired data. These include image-text contrastive learning [1–3], image-text matching [4–6], and masked modeling [4, 7, 8]. Following pre-training, shared conceptual representations enable a variety of downstream tasks, including text-to-image generation [2, 9], image captioning [6, 8], and vision-based question answering [5, 10]. Among these, one of the most popular multimodal models is Contrastive Language–Image Pre-training (CLIP) [1], which effectively learns visual concepts from language supervision through contrastive learning. CLIP jointly trains a vision model and a language model by embedding a large corpus of image-text pairs into a shared embedding space using self-supervised learning. The training process employs a contrastive learning objective [11], which encourages the model to bring similar pairs closer together in the embedding space, while it pushes dissimilar pairs further apart. The approach excels in zero-shot transfer for many downstream tasks across vision and language, even matching the performance of fully supervised models [1].

Despite their empirical success, a complete understanding of the mechanisms underlying multimodal learning models is lacking, with their behavior in certain scenarios even defying intuitive expectations about their design. For instance, while the contrastive loss is designed to align image embeddings with their corresponding text pairs, recent studies [12] have revealed a surprising phenomenon known as *modality gap*. As illustrated in Figure 1, this gap manifests as a significant separation between the embeddings of matched image-text pairs, with the two modalities occupying approximately parallel yet distant spaces [13]. Deepening our understanding of the factors underlying modality gap could shed light on the mechanisms driving the success of multimodal learning, as well as pave the way towards developing more effective multimodal models.

---

*The first two authors contributed to this work equally.

Second Conference on Parsimony and Learning (CPAL 2025).

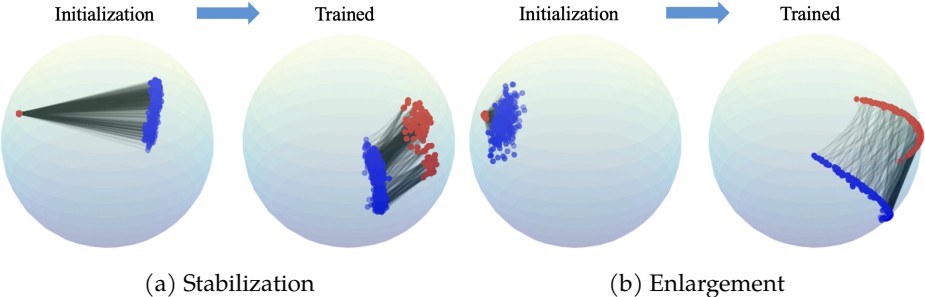

|  | Initialization | Trained | Initialization | Trained |

(a) Stabilization          (b) Enlargement

Figure 1: **Stabilization and enlargement of modality gap.** We visualize the CLIP text-image embedding space via PCA, where image features are in red and text features are in blue. A line connects each image-text pair. A modality gap emerges between image and text pairs. (a) After a long training, the gap between text and image still exists. (b) When two modalities are initialized with a small modality gap, the gap is still enlarged after training.

**Prior arts & limitations.** Driven by this goal, existing studies have investigated modality gap from a largely empirical perspective. However, a comprehensive and rigorous explanation of modality gap remains elusive. For example, Shi et al. [14] empirically demonstrated that modality gap could be caused by the types of initialization and temperature scaling of the loss on simple datasets, but they fall short of providing theoretical justifications of their studies. Schrodi et al. [15] investigated the role of imbalanced information between image and text modalities, implying that balancing the information complexity between text and image datasets could help mitigate modality gap. Again, their study is rather empirical without sufficient theoretical justifications. We postpone discussion of additional works and background on multimodal learning to Appendix A.

**Understanding modality gap through learning dynamics.** One surprising aspect of modality gap is that it contradicts with global optimality. Under ideal conditions, optimality conditions of the training loss imply perfect alignment between text and image embeddings. This is consistent with recent theoretical studies on neural collapse in classification problems [16], which demonstrate that, with sufficiently large model capacity and perfect training, the classification head and the embeddings become perfectly aligned [17–20]. Therefore, to gain deeper insight into the causes of modality gap, it is crucial to examine the factors influencing the learning dynamics of training these models. Moreover, empirical studies revealed several interesting phenomena in the learning dynamics, which could contribute to modality gap. Specifically, as shown by our experiments on both real datasets with practical networks (Figure 1) and synthetic datasets with simplified models (Figure 2), we observed the following when following the CLIP training procedure outlined by [1]:

- **Enlargement of modality gap under mismatch.** When there exist mismatches[2] of image and text embeddings at initialization, modality gap between image and text enlarges as training progresses, as shown in Figure 1b. In particular, as shown in Figure 2, this usually happens in the early phase of training when we start from random initializations, where modality gap first increases and then decreases after pairs of image and text embeddings become better aligned.

- **Stabilization of modality gap due to learned temperature.** Furthermore, as shown by [12], at random initialization, images and texts typically occupy distinct regions or "cones" within the feature space, resulting in a substantial modality gap. Observations in Figure 1a and Figure 2 imply that the gap-closing process remains limited when starting from random initialization. While the size of modality gap decreases in the later stages of training as mismatched pairs are reduced, it often stabilizes at a nonzero value, with limited reduction even as the training loss converges.

---

[2]For a pair $(\boldsymbol{h}_x, \boldsymbol{h}_y)$ of image and text embeddings, mismatch occurs if there exists an $\boldsymbol{h}'_x$ such that $\boldsymbol{h}'_x$ is "closer" to $\boldsymbol{h}_y$ than $\boldsymbol{h}_x$ (or the other way around)

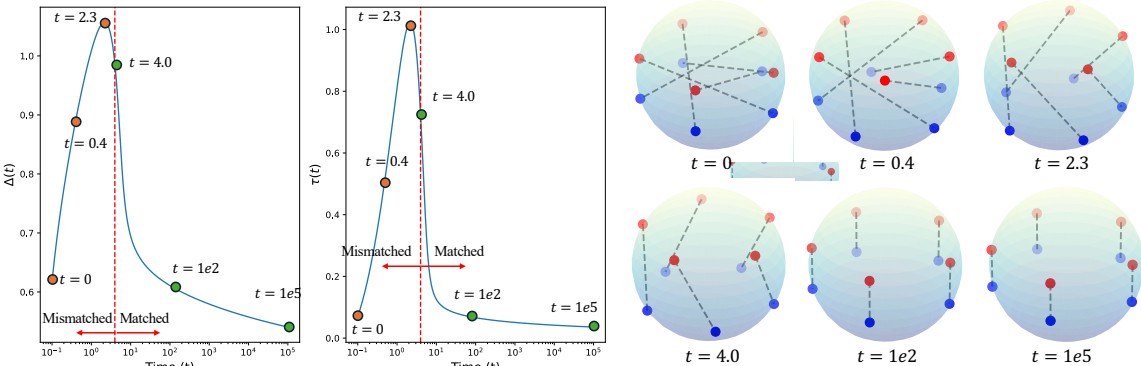

Figure 2: **Dynamics of modality gap $\Delta$ and temperature $\tau$ (defined in Section 2) during training on synthetic data.** Features from the two modalities are depicted in red and blue, with ground truth pairs connected by lines. At $t = 4.0$, all pairs are successfully matched. Initially, modality gap increases due to significant mismatches between pairs, but it decreases as the level of mismatch diminishes. Notably, modality gap and temperature exhibit highly coupled dynamics throughout the learning process.

**Summary of our contributions.** In this work, we conduct a theoretical analysis of contrastive multimodal learning by examining the learning dynamics through the lens of gradient flow – a largely overlooked yet critical perspective for understanding the cause of modality gap. Our approach reveals the factors contributing to modality gap, explaining why it may stabilize at a certain level or even increase during training. Based on these insights, we propose theory-guided approaches to reduce the gap and improve downstream performance. Our key contributions are as follows:

- **Theoretical contributions**. Through a careful gradient flow analysis, we demonstrate that modality gap diminishes at the extremely slow rate of $\Omega(1/\log(t)^2)$ in training time $t$, explaining why modality gap is prevalent in multi-modal models such as CLIP. Our analysis highlights the role of learned temperature in the persistence of modality gap, indicating several ways to reduce modality gap. Moreover, we rigorously demonstrate why and how modality gap can be created at initialization, suggesting that it cannot be simply closed before training.

- **Practical contributions.** Based on our theory, we propose practical methods, including temperature scheduling and exchanging features between modalities, to reduce modality gap and explore their benefits across several downstream tasks. We demonstrate that reducing the gap improves performance in image-text retrieval tasks but has a relatively smaller impact on visual classification including zero-shot and linear probing. In contrast, improving feature space uniformity proves to be more advantageous for visual classification tasks.

## 2. Problem Setup

In this section, we introduce the basic setup of the problem. We consider a set of $n$ paired training samples $\{(\boldsymbol{x}_i, \boldsymbol{y}_i)\}_{i=1}^n \subseteq \mathbb{R}^{d_x} \times \mathbb{R}^{d_y}$. Here, $(\boldsymbol{x}_i, \boldsymbol{y}_i)$ denotes a pair of two data points from different modalities (such as image and text) that are considered to be related to each other, e.g., $\boldsymbol{y}_i$ is the text caption of image $\boldsymbol{x}_i$. In multimodal learning, we want to align the image embedding $\boldsymbol{h}_{\boldsymbol{\theta},X}^i = f_{\boldsymbol{\theta}}(\boldsymbol{x}_i)$ and the text embedding of $\boldsymbol{h}_{\boldsymbol{\phi},Y}^i = g_{\boldsymbol{\phi}}(\boldsymbol{y}_i)$ through training two different deep networks $f_{\boldsymbol{\theta}} : \mathbb{R}^{d_x} \to \mathbb{R}^d$ and $g_{\boldsymbol{\phi}} : \mathbb{R}^{d_y} \to \mathbb{R}^d$ for each $i$.

**Contrastive loss for multimodal learning.** Let $\boldsymbol{H}_{\boldsymbol{\theta},X}, \boldsymbol{H}_{\boldsymbol{\phi},Y} \in \mathbb{R}^{n \times d}$ collectively denote the $\ell_2$-normalized embeddings of two different modalities:

$$\boldsymbol{H}_{\boldsymbol{\theta},X} = [\text{norm}(f_{\boldsymbol{\theta}}(\boldsymbol{x}_1)) \quad \dots \quad \text{norm}(f_{\boldsymbol{\theta}}(\boldsymbol{x}_n))]^\top, \quad \boldsymbol{H}_{\boldsymbol{\phi},Y} = [\text{norm}(g_{\boldsymbol{\phi}}(\boldsymbol{y}_1)) \quad \dots \quad \text{norm}(g_{\boldsymbol{\phi}}(\boldsymbol{y}_n))]^\top,$$

where the operator $\text{norm}(z) = z/\|z\|_2$ for any $z \in \mathbb{R}^d$. As shown in [1], we can jointly learn the parameters $\theta, \phi$ of networks $f, g$ respectively via

$$\min_{\theta, \phi, \nu} \ell(\beta(\nu) H_{\theta, X} H_{\phi, Y}^\top) \tag{1}$$

where $\ell : \mathbb{R}^{n \times n} \to \mathbb{R}$ is a certain contrastive loss. Here, following the convention in [1], $\beta(\cdot) : \mathbb{R} \to \mathbb{R}^+$ is the inverse temperature as a function of some learnable parameter $\nu$, i.e., we have $\beta(\nu) = 1/\tau(\nu) = \exp(\nu)$ for training CLIP models where $\tau$ is the conventional temperature. Additionally, for ease of exposition, we drop the superscripts and just write $H_X, H_Y$.

The contrastive loss $\ell$ is determined solely by the pairwise ($\beta$-scaled) inner products between modalities. Its objective is to maximize the diagonal entries of $\beta H_X H_Y^\top$ while minimizing the off-diagonal elements. For instance, CLIP achieves this using a specific *symmetric* cross-entropy (CE) loss for $\ell$, which we also adopt in this work. To derive this loss, we first define the standard (softmax) CE loss, $\ell_{\text{CE}}(m, e^{(i)})$, for logits $m \in \mathbb{R}^n$ for a target one-hot distribution $e^{(i)} \in \mathbb{R}^n$ as:

Figure 3: **Parallel modalities with or without mismatched pairs.** Ground truth pairs are connected by a dashed line.

$$\ell_{\text{CE}}(m, e^{(i)}) := \log \left( \sum_{j=1}^n \exp(m_j) \right) - m_i.$$

Then, we can introduce $\ell$ as applying the usual CE loss to each row and column of $M$ individually and then averaging them:

$$\ell(M) := \frac{1}{2n} \sum_{i=1}^n \left[ \ell_{\text{CE}}(M_{:,i}, e^{(i)}) + \ell_{\text{CE}}(M_{i,:}, e^{(i)}) \right]. \tag{2}$$

**Training loss with parallel embeddings.** Motivated by recent empirical studies [13], in this work we assume that the representations of each modality are *parallel*. Specifically, Zhang et al. [13] has empirically discovered that *inter-modality* variance and *intra-modality* variance are found to be orthogonal. Mathematically, we can impose the parallel constraint by replacing $H_X$ with $\tilde{H}_X = \begin{bmatrix} \sqrt{1 - \gamma_X^2} H_X & \gamma_X 1_n \end{bmatrix}$ and $H_Y$ with $\tilde{H}_Y = \begin{bmatrix} \sqrt{1 - \gamma_Y^2} H_Y & \gamma_Y 1_n \end{bmatrix}$, where $\gamma_X, \gamma_Y \in [-1, 1]$. For simplicity, we assume that $\gamma_X = \gamma$ and $\gamma_Y = -\gamma$ for some $\gamma \in [-1, 1]$. Then, (1) becomes

$$\min_{\theta, \phi, \nu, \gamma} \ell(\beta(\nu) \tilde{H}_X \tilde{H}_Y^\top) = \ell(\beta(\nu)(1 - \gamma^2) H_X H_Y^\top). \tag{3}$$

**Gradient flow dynamics.** We consider the training dynamics of (3) under continuous-time gradient flow, i.e., for time $t \geq 0$, the quantities $\theta(t), \phi(t), \nu(t), \gamma(t)$ satisfy

$$\frac{d\theta(t)}{dt} = -\frac{\partial \ell}{\partial \theta}, \quad \frac{d\phi(t)}{dt} = -\frac{\partial \ell}{\partial \phi}, \quad \frac{d\nu(t)}{dt} = -\frac{\partial \ell}{\partial \nu}, \quad \frac{d\gamma(t)}{dt} = -\frac{\partial \ell}{\partial \gamma} \tag{4}$$

with initial conditions $\theta(0) = \theta_0$, $\phi(0) = \phi_0$, $\nu(0) = \nu_0$, and $\gamma(0) = \gamma_0$. We refer the reader to Appendix B.1 for the derivative and gradient computations.

**Measures of modality gap and margin.** Based on the above assumptions of parallel embeddings between two modalities, we introduce two metrics of modality gap and margin that will be useful in our analysis. Given the reparameterized embeddings $\tilde{H}_X$ and $\tilde{H}_Y$ of the image $X$ and text $Y$ that we introduced above, where each row $\tilde{h}_x^i, \tilde{h}_y^i$ of $\tilde{H}_X, \tilde{H}_Y$ respectively correspond to a data sample, we define the modality centers as

$$c_X = \frac{1}{n} \sum_{j=1}^n \tilde{h}_x^j, \quad c_Y = \frac{1}{n} \sum_{j=1}^n \tilde{h}_y^j.$$

Following [12], we can measure the *modality gap* by the center distance between two modalities as

$$\Delta := \|\boldsymbol{c_X} - \boldsymbol{c_Y}\|.$$

As such, a larger center distance implies a larger modality gap. Moreover, the modality gap $\Delta$ also satisfies $\Delta \geq 2\gamma$, which will simplify our analysis in Section 3.

In our analysis, we also introduce a notion of *margin* to measure the match (angles) between associated pairs $(\boldsymbol{x}_i, \boldsymbol{y}_i)$ in the embedding space. Intuitively, we want to measure the matchness by measuring the correlation between $\boldsymbol{x}_i$ and $\boldsymbol{y}_i$ in the embedding space. Let $\boldsymbol{Z} = \boldsymbol{H}_X \boldsymbol{H}_Y^\top$, which compute the correlation between $\boldsymbol{H}_X$ and $\boldsymbol{H}_Y$. We define the margin by

$$\alpha(\boldsymbol{Z}) := \min_{i \neq j} (\boldsymbol{Z}_{i,i} - \boldsymbol{Z}_{i,j}) \wedge (\boldsymbol{Z}_{j,j} - \boldsymbol{Z}_{i,j}),$$

where $\wedge$ denotes minimum. Based on $\alpha(\boldsymbol{Z})$, we say that $\boldsymbol{H}_X, \boldsymbol{H}_Y$ are *perfectly matched* if $\alpha(\boldsymbol{H}_X \boldsymbol{H}_Y^\top) > 0$, otherwise we say that they are *mismatched*. An illustration of perfectly matched and mismatched data pairs is shown in Figure 3. Intuitively, perfectly matched means all ground truth pairs are the closest to each other.

## 3. Explaining the Modality Gap

In this section, we provide our main theoretical results that explain how modality gap emerges and remains throughout training.

### 3.1. Learning Temperature Stabilizes Modality Gap

As shown in Figure 2, modality gap $\Delta \geq 2\gamma$ and temperature $\tau$ are highly coupled, implying that learnable temperature plays a crucial role in the rate at which modality gap closes. Based upon the problem setup in Section 2, the following lemma directly reveals this relationship.

**Lemma 3.1.** *Let $\nu(t)$ and $\gamma(t)$ be solutions to the gradient flow dynamics given in* (4)*. Then we have*

$$R = \frac{d\gamma/dt}{d\beta/dt} = -\frac{2\beta(\nu)\gamma}{\beta'(\nu)^2 (1 - \gamma^2)} \tag{5}$$

*for all $t \geq 0$, where $\beta'(\nu)$ denotes the derivative of $\beta$ with respect to $\nu$. Moreover, given $\beta(\nu) = \exp(\nu)$, we have $\beta'(\nu) = \beta$ and the following holds:*

$$\beta = \beta_0 \sqrt{\frac{\gamma_0}{\gamma}} \exp\left(\frac{\gamma^2 - \gamma_0^2}{4}\right), \quad \text{and} \quad R = \Theta(1/\beta). \tag{6}$$

The proof of Lemma 3.1 is provided in Appendix B.2. The lemma relates the rates at which $\gamma$ and $\beta$ change, while $R = \Theta(1/\beta)$ implies that the ratio $R$ decays to zero as $\beta$ grows to infinity. This implies that an increasing $\beta$ will dominate the decrease in $\gamma$, preventing modality gap $\Delta$ from closing (i.e., its lower bound $2\gamma$ remains positive). This is made precise in the following theorem.

**Theorem 3.2.** *Based on the problem setup in Section 2, consider the gradient flow dynamics* (4) *for solving* (3)*. Suppose $\boldsymbol{Z}(t) = \boldsymbol{H}_X(t)\boldsymbol{H}_Y^\top(t)$ and the initial temperature $\beta_0 \geq \log(4(n-1))/(\overline{\alpha}(1 - \gamma_0^2))$ with $\beta(\nu) = \exp(\nu)$, and assume that the margin satisfies $\alpha(\boldsymbol{Z}(t)) \geq \overline{\alpha}$ for all $t \geq 0$ for some $\overline{\alpha} > 0$. Then modality gap $\Delta$ satisfies*

$$\Delta(t) \geq \Omega\left(\frac{1}{\log(t)^2}\right) \text{ for all } t \geq 0. \tag{7}$$

The proof can be found in Appendix B.3. To elaborate, consider the simplified setting where we replace $\ell$ in (3) with the scalar function $\ell(m) = \exp(-m)$ mimicking the exponential tail of the cross-entropy loss. The gradient flow in this case is simply given by $d\gamma/dt = -2\beta\gamma \exp(-\beta(1 - \gamma^2))$, so via the equality (6) we have $d\gamma/dt = \Omega\left(-\gamma^{1/2} \exp(-c\gamma^{-1/2})\right) = \Omega\left(-\gamma^{3/2} \exp(-c'\gamma^{-1/2})\right)$, for some $c' < c$. Integrating this equation and applying the inequality $\Delta \geq 2\gamma$ yields $\Delta(t) \geq \Omega(1/\log(t)^2)$.

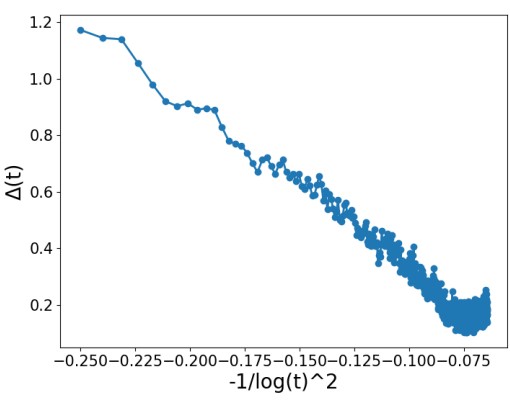

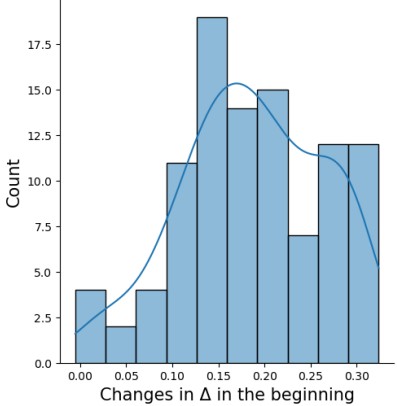

(a) Slow closure of modality gap

(b) Enlargement of modality gap

Figure 4: **Verifying theoretical results.** We sample 2048 random pairs of MSCOCO examples and utilize a standard CLIP model to (a) plot $\Delta$ throughout training from scratch and (b) plot a histogram of the change in $\Delta$ across 100 initializations in the first 10 steps of training. More details regarding the experimental setup can be found in Appendix E.

**Remarks.** We discuss Theorem 3.2 in the following:

- **Slow closure of modality gap.** The result in (7) indicates that $\Delta$, with rate $\Omega(1/\log(t)^2)$, approaches zero exceedingly slowly. Consequently, this lower bound implies that closing modality gap would require an impractically long training time. As shown in Figure 4a, this can be verified in practice on CLIP models trained on the MSCOCO dataset. Specifically, we sample 2048 random pairs of data and train the model from scratch for 10000 steps and record modality gap during training. In Figure 4a, we report modality gap from the 100th step when the gap begins to decrease consistently. The modality gap $\Delta(t)$ versus $-1/\log(t)^2$ exhibits a linear relationship, verifying our result and slow closure.

- **Discussion on the assumptions.** In Theorem 3.2, we assume a positive margin $\overline{\alpha} > 0$ throughout training, ensuring a minimum $\overline{\alpha}$ gap between perfectly matched and mismatched pairs. While primarily for analytical purposes, this assumption can be relaxed in practice, as shown in the early stage of Figure 4a, where the result holds even with many mismatched pairs. Additionally, while parallel embeddings between modalities are typically valid, breaking this constraint can help to mitigate modality gap, a strategy we leverage in Section 4. We also assume a sufficiently large initial inverse temperature $\beta_0$, a common practice [1]. Crucially, the choice $\beta(\nu) = \exp(\nu)$ is essential for achieving the rate in Theorem 3.2; changing $\beta$ or employing a temperature schedule (see Section 4) can significantly change the convergence rate of modality gap. For details on convergence rates with various schemes, see Appendix C. We evaluate these methods for reducing modality gap and their impact on downstream performance in Section 4.

## 3.2. Mismatched Pairs Enlarge Modality Gap at Early Training Stages

Second, we show that modality gap can be enlarged at the early stage of training, due to the large amount of mismatched pairs caused by random initialization. This further adds to the difficulty of closing modality gap.

**Theorem 3.3.** *Suppose the rows of $\boldsymbol{H}_X(0)$, $\boldsymbol{H}_Y(0)$ are drawn independently and uniformly from $\mathbb{S}^{d-1}$ and let $a = \beta_0(1 - \gamma_0^2)$. Then with probability $1 - \delta$, we have*

$$\left.\frac{d\Delta}{dt}\right|_{t=0} \geq 4\beta_0\gamma_0 \left( \frac{\xi_1 - 2e^a\epsilon}{\xi_2 + (e^a - e^{-a})\epsilon} - 2\epsilon \right) \tag{8}$$

*where $\epsilon = \sqrt{\log((4n+1)/\delta)/(2n)}$ and*

$$\xi_1 = \Gamma\left(\frac{d}{2}\right)\left(\frac{2}{a}\right)^\rho \left(I_{\rho-1}(a) - \frac{2\rho}{a}I_\rho(a)\right), \quad \xi_2 = \Gamma\left(\frac{d}{2}\right)\left(\frac{2}{a}\right)^\rho I_\rho(a),$$

*where $\rho = (d-2)/2$, $\Gamma$ is the gamma function, and $I_\rho(z)$ is the modified Bessel function of the first kind.*

The proof of Theorem 3.3 is given in Appendix B.4. In addition to the parallel modalities assumption, we assume that embeddings are uniformly distributed on the hypersphere. This aligns with practice, as the final-layer linear projections of $f_\theta$ and $g_\phi$ are typically initialized from a zero-mean isotropic Gaussian distribution, resulting in normalized features uniformly spread over the hypersphere. Next, we discuss Theorem 3.3 in the following:

- **Interpretation.** As the number of training samples $n$ is very large in practice, we can analyze the form of (8) in the limit $n \to \infty$, which gives

$$\left.\frac{d\Delta}{dt}\right|_{t=0} \geq 4\beta_0\gamma_0 \frac{\xi_1}{\xi_2} = 4\beta_0\gamma_0 \left(\frac{I_{\rho-1}(a)}{I_\rho(a)} - \frac{2\rho}{a}\right) \tag{9}$$

  almost surely. From the recurrence relation $I_{\rho-1}(a) = I_{\rho+1}(a) + (2\rho/a)I_\rho(a)$ in [21, (3.1.1)] and the fact that $I_\rho(z) > 0$ for real order $\rho$ and $z > 0$, we have

$$\frac{I_{\rho-1}(a)}{I_\rho(a)} - \frac{2\rho}{a} = \frac{I_{\rho+1}(a)}{I_\rho(a)} > 0$$

  so $d\Delta/dt > 0$ at $t = 0$, i.e., modality gap enlarges initially. Moreover, we can write (9) in terms of elementary functions in the special case $d = 3$. From $\rho = 1/2$, we have

$$\left.\frac{d\Delta}{dt}\right|_{t=0} \geq 4\beta_0\gamma_0 \left(\frac{I_{-1/2}(a)}{I_{1/2}(a)} - \frac{1}{a}\right) = 4\beta_0\gamma_0 \left(\frac{\cosh(\beta_0(1-\gamma_0^2))}{\sinh(\beta_0(1-\gamma_0^2))} - \frac{1}{\beta_0(1-\gamma_0^2)}\right)$$

  from closed-form expressions for half odd integer order Bessel functions [21, (3.3)]. For $\gamma_0 = \Theta(1)$, this gives $d\Delta/dt \geq \Theta(\beta_0 \tanh(\beta_0)) \sim \beta_0$. As such, a large initial inverse temperature (which is used in practice) encourages a large increase in $\Delta$ at initialization.

- **Experimental verification.** We verify that $d\Delta/dt > 0$ at initialization as implied by Theorem 3.2 in practice via randomly initializing CLIP models and recording the change in modality gap after the first 10 steps. We measure the change in modality gap with 100 independent trials and present the distribution of changes in Figure 4b. From Figure 4b, we can see modality gap increases at the start of training for all random initialization, with the mean being near $0.15$.

## 4. Mitigating the Modality Gap

The analysis of learning dynamics in Section 3 offers valuable insights into mitigating modality gap. It inspires us to design two types of methods for reducing modality gap: (*i*) **Temperature Control**, where we propose new temperature scheduling rules for accelerating the convergence rate of modality gap, and (*ii*) **Modality Swapping**, we proposed methods to manually break the parallel constraints of two modalities by swapping them during training. To evaluate these approaches, we train models from scratch on the MSCOCO dataset with different variants of the proposed methods and then measure modality gap and evaluate the performance on downstream tasks.In the following, we introduce the proposed methods in Section 4.1 and discuss the results and implications of reducing modality gap in Section 4.2.

### 4.1. Methods

We introduce the main idea of each method, and leave details on implementations to Appendix D.

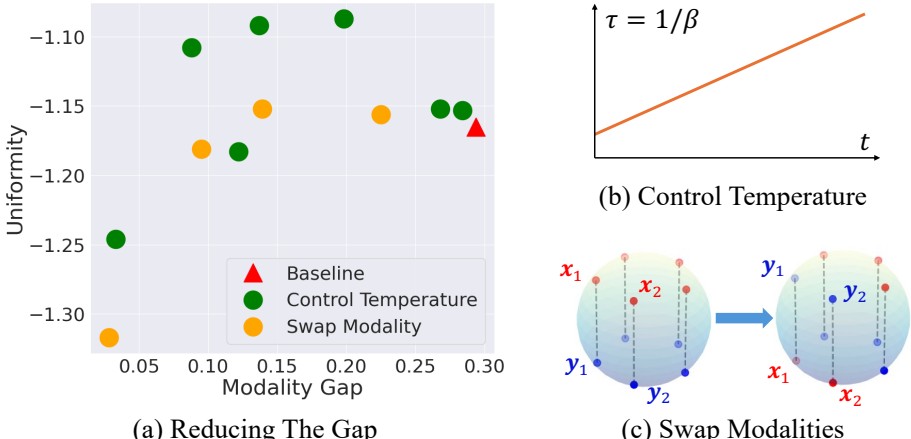

Figure 5: **We propose two categories of methods: Control Temperature and Swap Modality.** (a) Our methods reduce modality gap and may influence the uniformity of the feature space. (b) **Control Temperature** maintains the temperature at larger values such as increasing temperature across training. (c) **Swap Modalities** swaps between image and text feature pairs.

**Temperature Control.** Our first type of method involves the control of temperature $\tau(\nu)$ during training. As defined in (1), $\beta(\nu) = 1/\tau(\nu) = \exp(\nu)$ is the temperature parameterization. Usually, $\tau(\nu)$ is the so-called *temperature* in CLIP loss [1, 11]. Thus, we refer to $\tau(\nu)$ as *temperature* when we introduce the following methods. As shown by analysis in Section 3.1 and Appendix C, the learnable temperature parameter is the key factor causing the stabilization of the modality gap. More specifically, the fast decrease in temperature outpaces the decrease in the modality gap, preventing it from closing. The following methods all counteract this decrease:

- Temperature Scheduling (TS): Instead of allowing $\tau(\nu)$ to diminish freely with $\nu$ (as shown in Figure 5(b)), we enforce a linearly increasing schedule for $\tau$ during training.

- Temperature Reparameterization (TR): As mentioned in Section 3.1, the specific parameterization of temperature also affects the gap closing rate. We replace the original parameterization, $1/\tau(\nu) = \beta(\nu) = \exp(\nu)$, with alternatives that yield a higher gap closing rate, such as $1/\tau(\nu) = \log(1 + e^{\nu})$.

- Smaller Learning Rate of Temperature (SLRT): We use a smaller learning rate for the temperature parameter compared to other learnable parameters. This slows down the decrease in temperature, allowing larger temperature values to be maintained during training.

- Fixed on Large Temperature (FLT): Rather than allowing the temperature to be freely learned, we fix $\tau$ at a high value throughout the training process.

**Modality Swapping.** Our second approach involves manually mixing the two modalities by swapping pairs, as illustrated in Figure 5c. This mixing prevents the two modalities from remaining segregated into parallel planes. Consequently, the repulsion caused by mismatched pairs no longer occurs between the original planes, thereby reducing the overall repulsion between them. We introduce two strategies for swapping pairs—hard swapping and soft swapping—both of which effectively mitigate modality gap.

- Hard Swapping Between Modalities (HS). During training, we randomly select some images and their paired text descriptions. Then, we exchange the features of these images and the paired texts in the shared feature space, as visualized in Figure 5c.

- Soft Swapping Between Modalities (SS). Different from hard swapping, for a pair of image and text features, soft swapping mixes the two features to create a pair of image and text features.

## 4.2. Experimental Results and Implications

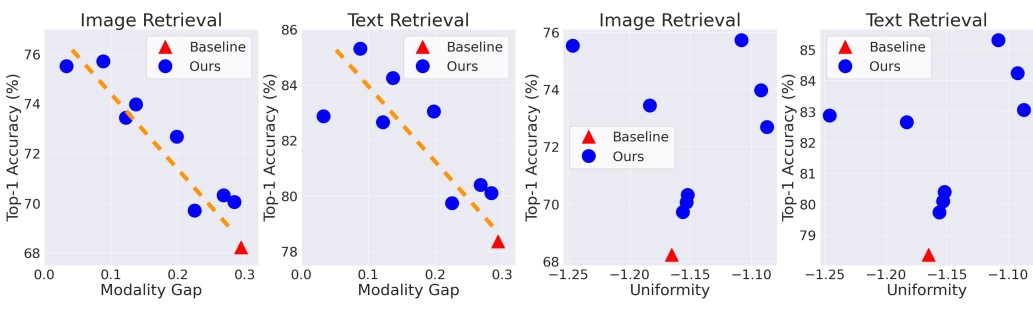

(a) Retrieval accuracies vs. modality gap.  (b) Retrieval accuracies vs. uniformity.

Figure 6: **Image-text retrieval accuracy increases with reduced modality gap by our proposed methods.** Uniformity does not show strong correlations with retrieval accuracies.

**Experimental setup.** We train CLIP models from scratch on MSCOCO using the proposed methods and the original CLIP training process as a baseline. After pretraining, we evaluate two key attributes of the shared feature space: (*i*) modality gap between image and text features and (*ii*) feature space uniformity. Additionally, we assess the models on four downstream tasks: (*i*) zero-shot image classification, (*ii*) linear-probe image classification, (*iii*) image-text retrieval, and (*iv*) vision-language question answering (MMVP-VLM [22]). For each training setting, we independently train three models and report averaged results for attributes and task performance. Detailed experimental setup is provided in Appendix E.

**Closing modality gap is especially helpful for image-text retrievals.** We visualize the correlation between image-text retrieval accuracies and modality gap in Figure 6a, and the correlation between image-text retrieval accuracies and uniformity in Figure 6b. As shown in the figure, though these methods are different variants of Control Temperature and Swap Modality, a smaller modality gap clearly leads to a higher retrieval accuracy. This indicates reducing modality gap improves image-text retrieval. In contrast, uniformity does not show a strong correlation with the retrieval accuracy. We include more detailed results for each method variant in Appendix F.

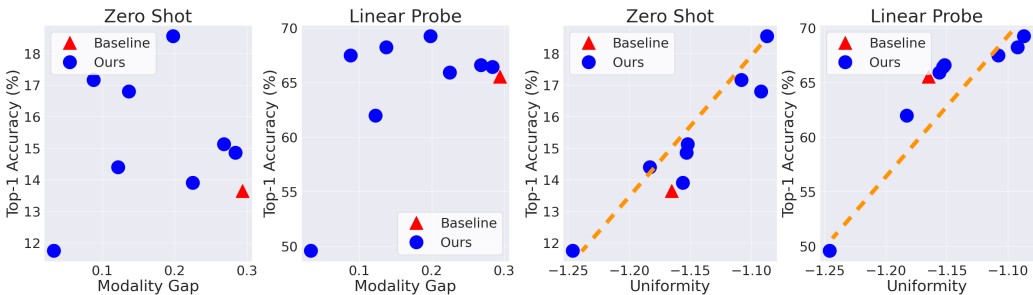

(a) Zero shot/linear probe vs. modality gap.  (b) Zero shot/linear probe vs. uniformity.

Figure 7: **Zero-shot and linear probe accuracies are not strongly correlated with modality gap.** Uniformity has a clear positive correlation with zero-shot and linear probe accuracies instead.

**Closing modality gap is not all you need.** Even though methods reducing modality gap increase retrieval performance, it does not mean modality gap is the only metric we should care about when characterizing the space. For example, as shown in Figure 7, bad uniformity will harm zero-shot and linear probe accuracies while a reduced modality gap does not have an obvious effect on them. Moreover, for the difficult vision-language question-answering task MMVP-VLM, we test different variants of Fixed on Large Temperature (FLT) and Temperature Scheduling (TS), and neither

| | Baseline | FLT | | | TS | | |
|---|---|---|---|---|---|---|---|
| | | $4 \cdot 10^{-2}$ | $7 \cdot 10^{-2}$ | $10^{-1}$ | S1 | S2 | S3 |
| Modality Gap ($\downarrow$) | 0.294 | 0.122 | 0.033 | **0.019** | 0.198 | 0.137 | 0.088 |
| Uniformity ($\uparrow$) | -1.165 | -1.183 | -1.246 | -1.251 | **-1.087** | -1.092 | -1.108 |
| MMVP-VLM ($\uparrow$) | 14.321 | 11.852 | 14.074 | **14.817** | 14.074 | 14.321 | 13.087 |

Table 1: **MMVP-VLM accuracy does not strongly correlate with modality gap or uniformity.**

modality gap nor uniformity shows a strong correlation with the MMVP-VLM performance. We discuss the detailed settings of different variants in Appendix D and Appendix F

## 5. Conclusion

In this work, we investigated the modality gap in training multimodal models. By analyzing gradient flow learning dynamics, we theoretically characterized how learning temperature and mismatched pairs influence this gap. Based on our analysis, we proposed principled methods to control temperature and swap information between modalities to reduce the gap, which also improves downstream task performance—particularly in retrieval tasks. Our work opens up future directions, such as extending gradient flow analysis to study the difficulty of closing the gap with varying levels of shared information between modalities by modeling data distributions. Additionally, our results can provide insights on finetuning scenarios where domain differences between pretraining and finetuning data need to be considered.

## Acknowledgment

We acknowledge support from NSF CAREER CCF-2143904, NSF IIS 2312842, NSF IIS 2402950, and ONR N00014-22-1-2529. Additionally, we acknowledge fruitful discussions with Yuexiang Zhai (UC Berkeley) and Liyue Shen (UMich).

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

# A. Related Works

**Contrastive Learning**  In the past, contrastive learning has been demonstrated as a powerful tool to learn reasonable representations in a self-supervised way [11, 23–25]. Representative contrastive methods including SimCLR [11], MOCO [24], and SwAV [25]. Intuitively, contrastive learning aims to pair together data that contain similar information, and separate pairs that contain distinct information. Such a strategy can effectively extract key features within data that are useful for various downstream tasks. For example, by utilizing contrastive learning in images or videos, we can get image or video encoders that can perform zero-shot image and video retrieval, as well as image classification [11, 24, 25], action detection, action segmantation, and other tasks [26, 27]. Moreover, contrastive learning can be applied to a wide range of data, from language to protein [28, 29]. Apart from application, other works approach to analyze and understand contrastive learning from the perspective of alignment & uniformity [30, 31], specific design choices [32, 33], and so on [34, 35].

**Multimodal Learning**  Multimodal learning aims to learn a shared representation space for different modalities such as images and texts, where the learned representation space can express the shared information hidden in different modalities [1, 36, 37]. Under this context, contrastive learning has been proved applicable and powerful to multimodal learning [1]. The contrastive loss is applied between two modalities (take image and text as an example) such that positive pairs (i.e., image and text pairs that contain similar information) have similar representations but negative pairs (i.e., image and text pairs that contain distinct information) are apart from each other. The pretrained multimodal models excel in various downstream tasks such as linear probe, zero-shot classification, and retrieval [1], and are even useful for medical applications [38] and generative models [39–41]. Despite its great power, recently, several works have also identified bias and flaws in the learned multimodal models, such as uni-model bias [42], and failure in understanding detailed information [22], which add difficulty to applications such as text-to-image editing [43, 44]. Therefore, the theoretical understanding of contrastive multimodal learning is an important problem for further solving the aforementioned problems.

**Modality Gap**  Following prior discoveries, existing works delve deeper into the modality gap from both theoretical and empirical perspectives. For instance, [14] experimentally demonstrates the influence of different initialization and temperature parameters in the modality gap with toy datasets, but stops short of providing theoretical justifications for the emergence of modality gap under various conditions. [15] investigates the role of information imbalance between modalities, suggesting that balancing information between text and image datasets helps close the modality gap. However, their claim is based solely on experiments with toy datasets and lacks a strong theoretical foundation. Instead of focusing on the reasons behind the modality gap, other studies propose practical solutions. For example, [45] and [46] introduce new loss functions for fine-tuning the CLIP model to close the gap, while [47] encourages different modalities to share portions of their encoders. Although effective, they remain largely experimental and fail to provide a rigorous explanation for modality gap's origin.

# B. Proofs

## B.1. Preliminaries

The gradient of $\ell$ given in (2) is given by

$$\nabla\ell(\boldsymbol{M}) = \frac{1}{2n}\left(\frac{\exp(\boldsymbol{M})}{\mathbf{1}_n\mathbf{1}_n^\top\exp(\boldsymbol{M})} + \frac{\exp(\boldsymbol{M})}{\exp(\boldsymbol{M})\mathbf{1}_n\mathbf{1}_n^\top}\right) - \frac{1}{n}\boldsymbol{I}_n$$

where division and exponentiation are carried out element-wise.

Define the function $g_a(\boldsymbol{Z}) := -\langle \boldsymbol{Z}, \nabla \ell(a\boldsymbol{Z}) \rangle$ which can be written as

$$g_a(\boldsymbol{Z}) = \frac{1}{2n} \left( \sum_{i=1}^{n} \mu_i(\boldsymbol{Z}_{:,i}, a) + \mu_i(\boldsymbol{Z}_{i,:}, a) \right)$$

where

$$\mu_i(\boldsymbol{z}, a) = \langle \boldsymbol{z}, \boldsymbol{e}^{(i)} - \sigma(a\boldsymbol{z}) \rangle \tag{10}$$

and $\sigma$ is the softmax function $\sigma(\boldsymbol{m}) = \exp(\boldsymbol{m})/\sum_j \exp(\boldsymbol{m}_j)$. Then we have

$$\frac{\partial \ell}{\partial \nu} = -\beta'(\nu)(1-\gamma^2)g_{\beta(\nu)(1-\gamma^2)}(\boldsymbol{Z}), \quad \frac{\partial \ell}{\partial \gamma} = 2\beta(\nu)\gamma g_{\beta(\nu)(1-\gamma^2)}(\boldsymbol{Z})$$

which gives the gradient flow dynamics

$$\frac{d\beta}{dt} = (\beta')^2(1-\gamma^2)g_{\beta(1-\gamma^2)}(\boldsymbol{Z}), \quad \frac{d\gamma}{dt} = -2\beta\gamma g_{\beta(1-\gamma^2)}(\boldsymbol{Z}) \tag{11}$$

where $\beta' := \beta'(\nu)$.

## B.2. Proof of Lemma 3.1

*Proof.* From (11), we have

$$\frac{1}{2\beta\gamma}\left(-\frac{d\gamma}{dt}\right) = \frac{1}{(\beta')^2(1-\gamma^2)}\frac{d\beta}{dt} \implies R = \frac{d\gamma/dt}{d\beta/dt} = -\frac{2\beta\gamma}{(\beta')^2(1-\gamma^2)}$$

which gives (5). Now let $\beta = \exp(\nu)$ so $\beta' = \beta$. Then we can rewrite and integrate as

$$\int_{\beta_0}^{\beta} \frac{d\beta}{\beta} = -\frac{1}{2}\int_{\gamma_0}^{\gamma} \left(\frac{1}{\gamma} - \gamma\right) d\gamma \implies \beta(\gamma) = \beta_0\sqrt{\frac{\gamma_0}{\gamma}}\exp\left(\frac{\gamma^2 - \gamma_0^2}{4}\right)$$

which gives (6). $\qquad\square$

## B.3. Proof of Theorem 3.2

**Lemma B.1.** *Let $\boldsymbol{z} \in \mathbb{R}^n$ and $i \in [n]$.*

(a) *If $\boldsymbol{z}_i \geq \max_j \boldsymbol{z}_j$, then $\mu_i(\boldsymbol{z}, a) \geq 0$ for all $a \geq 0$.*

(b) *For any $\alpha > 0$, we have*

$$\mu_i(\boldsymbol{z}, a) \leq \mu_i(\alpha(\boldsymbol{e}^{(i)} - \mathbf{1}), a)$$

*for all $a \geq \log(4n-4)/\alpha$ and for all $\boldsymbol{z}$ such that $\boldsymbol{z}_i - \max_{j\neq i} \boldsymbol{z}_j \geq \alpha$.*

*Proof.* We have

$$\langle \boldsymbol{z}, \boldsymbol{e}^{(i)} \rangle = \boldsymbol{z}_i \geq \max_j \boldsymbol{z}_j = \sum_l \sigma_l(a\boldsymbol{z})\left(\max_j \boldsymbol{z}_j\right) \geq \sum_l \sigma_l(a\boldsymbol{z})\boldsymbol{z}_l = \langle \boldsymbol{z}, \sigma(a\boldsymbol{z}) \rangle$$

so $\mu_i(\boldsymbol{z}, a) \geq 0$, giving (a).

To prove (b), without loss of generality, we can take $i = 1$ and assume $\boldsymbol{z}_1 = 0$ and $\boldsymbol{z}_j \leq -\alpha$ for $j \neq 1$. Define $\boldsymbol{w} = \boldsymbol{z}_{2:n} \in \mathbb{R}^{n-1}$ and

$$\psi(\boldsymbol{w}, a) := \frac{\sum_j \boldsymbol{w}_j \exp(a\boldsymbol{w}_j)}{1 + \sum_j \exp(a\boldsymbol{w}_j)}$$

so that $\mu_1(\boldsymbol{z}, a) = -\psi(\boldsymbol{w}, a)$. The statement is then equivalent to showing $\psi(\boldsymbol{w}, a) \geq \psi(-\alpha\mathbf{1}, a)$ for any $\boldsymbol{w}$ such that $\boldsymbol{w}_j \leq -\alpha$ for all $j \in [n-1]$. It suffices to show that for any $k$, we have $\partial\psi/\partial\boldsymbol{w}_k \leq 0$

whenever $\boldsymbol{w} \preceq -\alpha \mathbf{1}$, i.e., $\psi$ is decreasing in any argument for the region bounded above by $-\alpha \mathbf{1}$. We compute

$$\frac{\partial \psi}{\partial \boldsymbol{w}_k} = \frac{\exp(a\boldsymbol{w}_k)}{1 + \sum_j \exp(a\boldsymbol{w}_j)} \left(1 + a\boldsymbol{w}_k - a\psi(\boldsymbol{w}, a)\right)$$

where

$$a\psi(\boldsymbol{w}, a) = \frac{\sum_j a\boldsymbol{w}_j \exp(a\boldsymbol{w}_j)}{1 + \sum_j \exp(a\boldsymbol{w}_j)} \geq \sum_j a\boldsymbol{w}_j \exp(a\boldsymbol{w}_j) \geq -(n-1)a\alpha \exp(-a\alpha)$$

provided that $\boldsymbol{w} \preceq -\alpha \mathbf{1}$, where the last inequality follows from $\alpha \geq 1/a$. Now, for any $x \geq \log(4n - 4)$, we have that $(n-1) \leq \exp(x)(1-1/x)$ by $1 - 1/x \geq 1/4$. Therefore, $x = a\alpha \geq \log(4n-4)$ satisfies

$$(n-1)\exp(-a\alpha) \leq 1 - \frac{1}{a\alpha} \implies -(n-1)a\alpha \exp(-a\alpha) \geq 1 - a\alpha.$$

Combining the above inequalities along with $1 + a\boldsymbol{w}_k \leq 1 - a\alpha$ yields $a\psi(\boldsymbol{w}, a) \geq 1 + a\boldsymbol{w}_k$, and thus $\partial\psi/\partial\boldsymbol{w}_k \leq 0$, completing the proof. $\qquad\square$

*Proof of Theorem 3.2.* First, we note by (11) that $\beta(t)$ is increasing in $t$ and $\gamma(t)$ is decreasing in $t$ due to the fact that $g_{\beta(1-\gamma^2)}(\boldsymbol{Z}) \geq 0$ via perfect mismatch and (a) in Lemma B.1. Now, we can substitute the expression for $\beta$ in (6) into the dynamics of $\gamma$ in (11) giving

$$\frac{d\gamma}{dt} = -2\beta(\gamma)\gamma g_{\beta(\gamma)(1-\gamma^2)}(\boldsymbol{Z}).$$

Since

$$\beta(\gamma)(1-\gamma^2) \geq \beta_0 \exp(-\gamma_0^2/4)(1-\gamma_0^2)$$

we have by $\beta_0 \geq \log(4n-4)/(\overline{\alpha}(1-\gamma_0^2))$, (b) in Lemma B.1, and the fact that $\beta(t)$ and $\gamma(t)$ are increasing and decreasing in $t$ respectively that

$$g_{\beta(\gamma)(1-\gamma^2)}(\boldsymbol{Z}) \leq \mu_1\left(\overline{\alpha}(\boldsymbol{e}^{(1)} - \mathbf{1}), \beta(\gamma)(1-\gamma^2)\right)$$
$$= 1 - (p_1 + (1-\overline{\alpha})(1-p_1)) = \overline{\alpha}(1-p_1)$$

where $p_1$ is the first entry of $\sigma\left(\beta(\gamma)(1-\gamma^2)\overline{\alpha}(\boldsymbol{e}^{(1)} - \mathbf{1})\right)$. We compute

$$1 - p_1 = \frac{1}{1 + \exp(\beta(\gamma)(1-\gamma^2)\overline{\alpha})/(n-1)} \leq (n-1)\exp(-\beta(\gamma)(1-\gamma^2)\overline{\alpha})$$

so overall we have

$$2\beta(\gamma)\gamma g_{\beta(1-\gamma^2)}(\boldsymbol{Z}) \leq 2\overline{\alpha}(n-1)\beta(\gamma)\gamma \exp(-\beta(\gamma)(1-\gamma^2)\overline{\alpha})$$
$$\leq 2\overline{\alpha}(n-1)\beta_0\sqrt{\frac{\gamma_0}{\gamma}}\gamma \exp\left(-\beta_0\sqrt{\frac{\gamma_0}{\gamma}}\exp\left(-\frac{\gamma_0^2}{4}\right)(1-\gamma_0^2)\overline{\alpha}\right)$$

where the second inequality follows from $\gamma \leq \gamma_0$ and $\exp\left(-\gamma_0^2/4\right) \leq \exp\left((\gamma^2 - \gamma_0^2)/4\right) \leq 1$. Let $c_1 = 2\overline{\alpha}(n-1)\beta_0\sqrt{\gamma_0}$ and $c_2 = \overline{\alpha}\beta_0\sqrt{\gamma_0}(1-\gamma_0^2)\exp(-\gamma_0^2/4)$, and define a new problem

$$\frac{d\tilde{\gamma}}{dt} = -c_1\sqrt{\tilde{\gamma}}\exp\left(-\frac{c_2}{\sqrt{\tilde{\gamma}}}\right)$$

with initial condition $\tilde{\gamma}(0) = \gamma_0$. Since $d\tilde{\gamma}/dt \leq d\gamma/dt$ whenever $\tilde{\gamma} = \gamma$, we have that $\tilde{\gamma}(t) \leq \gamma(t)$ for all $t \geq 0$. We make the change of variables $\omega = 1/\sqrt{\tilde{\gamma}}$ which gives

$$\frac{d\omega}{dt} = (1/2)c_1\omega^2\exp(-c_2\omega)$$

with $\omega(0) = 1/\sqrt{\gamma_0}$. From the fact that $(1/55)x^2\exp(-x/10) \leq 1$ for all $x \geq 0$, we have that

$$(1/2)c_1\omega^2\exp(-c_2\omega) \leq c_3\exp(-c_4\omega)$$

where $c_3 = 55c_1/2$ and $c_4 = c_2 - 1/10$ so defining

$$\frac{d\tilde{\omega}}{dt} = c_3 \exp(-c_4\tilde{\omega})$$

with $\tilde{\omega}(0) = \omega(0) = 1/\sqrt{\gamma_0}$ satisfies $\tilde{\omega}(t) \geq \omega(t)$ for all $t \geq 0$. Integrating, we have $\tilde{\omega}(t) = (1/c_4)\log(c_3 t + \exp(c_4/\sqrt{\gamma_0}))$, and combining all the above bounds yields

$$\gamma(t) \geq \frac{c_2}{\log(c_3 t + \exp(c_4/\sqrt{\gamma_0}))^2}$$

for all $t \geq 0$, which combined with $\Delta \geq 2\gamma$ completes the proof. $\qquad\square$

## B.4. Proof of Theorem 3.3

**Lemma B.2.** *Let $x, y \overset{iid}{\sim} Uniform(\mathbb{S}^{d-1})$. Then $z = \langle x, y \rangle \sim f_Z$ where*

$$f_Z(z) = \Gamma(d/2)/(\sqrt{\pi}\,\Gamma((d-1)/2))(1-z^2)^{(d-3)/2}, \quad z \in [-1,1]. \tag{12}$$

*Proof.* By symmetry, we can fix $y = e^{(1)}$. We have that $z^2 = x_1^2 \overset{d}{=} w_1^2/(w_1^2 + \cdots + w_d^2)$ where $w_1, \ldots, w_d \overset{iid}{\sim} \mathcal{N}(0,1)$. Letting $v_1 = w_1^2$ and $v_2 = w_2^2 + \cdots + w_d^2$ such that $v_1 \sim \chi_1^2$ and $v_2 \sim \chi_{d-1}^2$, we have $z^2 = v_1/(v_1 + v_2) \sim \text{Beta}(1/2, (d-1)/2)$ (see the relationship between chi-squared distribution and beta distribution in [48]). Then we have

$$F_{Z^2}(z^2) = \mathbb{P}\{Z^2 \leq z^2\} = \mathbb{P}\{-z \leq Z \leq z\} = 2F_Z(z)$$

hence

$$f_Z(z) = z f_{Z^2}(z^2) = \frac{\Gamma(d/2)}{\Gamma(1/2)\Gamma((d-1)/2)} z \cdot z^{2(1/2-1)}(1-z^2)^{(d-1)/2-1}$$

$$= \frac{\Gamma(d/2)}{\sqrt{\pi}\,\Gamma((d-1)/2)}(1-z^2)^{(d-3)/2},$$

completing the proof. $\qquad\square$

**Lemma B.3.** *Let $z \sim f_Z$ where $f_Z$ is given in (12) and $a > 0$. Then we have $\mathbb{E}[z] = 0$, and*

$$\mathbb{E}[\exp(az)] = \Gamma\left(\frac{d}{2}\right)\left(\frac{2}{a}\right)^\rho I_\rho(a) \tag{13}$$

$$\mathbb{E}[z\exp(az)] = \Gamma\left(\frac{d}{2}\right)\left(\frac{2}{a}\right)^\rho \left(I_{\rho-1}(a) - \frac{2\rho}{a}I_\rho(a)\right) \tag{14}$$

*where $\rho = (d-2)/2$, $\Gamma$ is the gamma function and $I_\rho(z)$ is the modified Bessel function of the first kind.*

*Proof.* The first claim $\mathbb{E}[z] = 0$ follows directly from symmetry of $f_Z$. Now define

$$F(a,d) = \frac{1}{\sqrt{\pi}\,\Gamma((d-1)/2)}\int_{-1}^{1}\exp(az)(1-z^2)^{(d-3)/2}\,dz.$$

From the identity in [49, (10.32.2)], we have that $F(a,d) = (2/a)^{(d-2)/2}I_{(d-2)/2}$. Now

$$\mathbb{E}[\exp(az)] = \frac{\Gamma(d/2)}{\sqrt{\pi}\,\Gamma((d-1)/2)}\int_{-1}^{1}\exp(az)(1-z^2)^{(d-3)/2}\,dz = \Gamma(d/2)F(a,d)$$

which gives (13). On the other hand, we have

$$\mathbb{E}[z\exp(az)] = \frac{d\,\mathbb{E}[\exp(az)]}{da} = \Gamma(d/2)\frac{dF(a,d)}{da}$$

where

$$\frac{dF(a,d)}{da} = \frac{1}{2}(1-d/2)\left(\frac{2}{a}\right)^{d/2}I_{(d-2)/2}(a) + \left(\frac{2}{a}\right)^{(d-2)/2}\frac{dI_{(d-2)/2}(a)}{da}$$

with
$$\frac{dI_{(d-2)/2}(a)}{da} = I_{(d-4)/2} - \frac{(d-2)}{2a}I_{(d-2)/2}(a)$$
by the identity in [21, (3.1.1)]. Putting together the above formulas and simplifying yields (14), completing the proof. □

*Proof of Theorem 3.3.* Let $\boldsymbol{Z} = \boldsymbol{H}_X(0)\boldsymbol{H}_Y^\top(0)$, so by Lemma B.2 we have $\boldsymbol{Z}_{ij} \sim f_Z$ where $f_Z$ is given by (12). We also note that any given row, column, or diagonal of $\boldsymbol{Z}$ forms a collection of independent variables (but not the entire matrix).

Let $\boldsymbol{z} \in [-1,1]^n$ be a given row, column, or the diagonal of $\boldsymbol{Z}$. From Lemma B.3 we have $\mathbb{E}[z] = 0$, and let $\xi_1 = \mathbb{E}[z \exp(az)]$ and $\xi_2 = \mathbb{E}[\exp(az)]$. By $\boldsymbol{z}_i \in [-1,1]$, $\boldsymbol{z}_i \exp(a\boldsymbol{z}_i) \in [-e^a, e^a]$, and $\exp(a\boldsymbol{z}_i) \in [e^{-a}, e^a]$ for all $i \in [n]$, applying one-sided Hoeffding inequalities gives

$$\mathbb{P}\left\{\frac{1}{n}\sum_j \boldsymbol{z}_j \leq 2\sqrt{\frac{\log(1/\delta)}{2n}}\right\} \geq 1 - \delta,$$

$$\mathbb{P}\left\{\frac{1}{n}\sum_j \boldsymbol{z}_j \exp(a\boldsymbol{z}_j) \geq \xi_1 - 2e^a\sqrt{\frac{\log(1/\delta)}{2n}}\right\} \geq 1 - \delta,$$

$$\mathbb{P}\left\{\frac{1}{n}\sum_j \exp(a\boldsymbol{z}_j) \leq \xi_2 + (e^a - e^{-a})\sqrt{\frac{\log(1/\delta)}{2n}}\right\} \geq 1 - \delta.$$

Let $a_0 = \beta_0(1 - \gamma_0^2)$. We compute

$$\left.\frac{d\gamma}{dt}\right|_{t=0} = -2\beta_0\gamma_0 g_{\beta_0(1-\gamma_0^2)}(\boldsymbol{Z})$$

$$= -\frac{\beta_0\gamma_0}{n}\left(\sum_{i=1}^n \mu_i(\boldsymbol{Z}_{:,i}, a_0) + \mu_i(\boldsymbol{Z}_{i,:}, a_0)\right)$$

$$= -\frac{\beta_0\gamma_0}{n}\left(\sum_{i=1}^n \langle \boldsymbol{Z}_{:,i}, \boldsymbol{e}^{(i)} - \sigma(a_0\boldsymbol{Z}_{:,i})\rangle + \langle \boldsymbol{Z}_{i,:}, \boldsymbol{e}^{(i)} - \sigma(a_0\boldsymbol{Z}_{i,:})\rangle\right)$$

$$= \frac{\beta_0\gamma_0}{n}\left(\sum_{i=1}^n \langle \boldsymbol{Z}_{:,i}, \sigma(a_0\boldsymbol{Z}_{:,i})\rangle + \sum_{i=1}^n \langle \boldsymbol{Z}_{i,:}, \sigma(a_0\boldsymbol{Z}_{i,:})\rangle - 2\sum_{j=1}^n \boldsymbol{Z}_{j,j}\right)$$

$$= \frac{\beta_0\gamma_0}{n}\left(\sum_{i=1}^n \frac{\frac{1}{n}\sum_{j=1}^n \boldsymbol{Z}_{j,i}\exp(a_0\boldsymbol{Z}_{j,i})}{\frac{1}{n}\sum_{j=1}^n \exp(a_0\boldsymbol{Z}_{j,i})} + \sum_{i=1}^n \frac{\frac{1}{n}\sum_{j=1}^n \boldsymbol{Z}_{i,j}\exp(a_0\boldsymbol{Z}_{i,j})}{\frac{1}{n}\sum_{j=1}^n \exp(a_0\boldsymbol{Z}_{i,j})} - 2\sum_{j=1}^n \boldsymbol{Z}_{j,j}\right).$$

Applying the above concentration inequalities yields

$$\left.\frac{d\gamma}{dt}\right|_{t=0} \geq 2\beta_0\gamma_0\left(\frac{\xi_1 - 2e^{a_0}\epsilon}{\xi_2 + (e^{a_0} - e^{-a_0})\epsilon} - 2\epsilon\right)$$

with probability $1 - \delta$ where $\epsilon = \sqrt{\log((4n+1)/\delta)/(2n)}$ by union bound for $4n+1$ events. Finally, we have $d\Delta/dt \geq 2d\gamma/dt$ by $\Delta \geq 2\gamma$ which gives the result. □

# C. Alternate Temperature Schemes

Consider the simplified setting $\ell(m) = \exp(-m)$ discussed in Section 3.

**Temperature reparameterization.** Take $\beta(\nu) = \nu$ with $\beta'(\nu) = 1$ for $\nu > 0$. Integrating (5) gives $\beta(\gamma) = \Theta\left(\log(1/\gamma)^{1/2}\right)$, which grows significantly slower compared to (6) as $\gamma \to 0$. As a result, the dynamics of $\gamma$ become $d\gamma/dt = \Theta\left(-\log(1/\gamma)^{1/2}\gamma\exp(-\log(1/\gamma)^{1/2})\right)$, which has solution $\gamma(t) = \Theta(1/t^{\log(t)})$, a significantly *faster* rate of convergence than the one in Theorem 3.2.

**Temperature scheduling.** Alternatively, one can consider a temperature *schedule*, where $\beta(t)$ is simply a function of time $t$. Then $\gamma(t) = \Theta\left(\exp\left(-\int_0^t \beta(s)\exp(-\beta(s))\,ds\right)\right)$. In the simplest case, we can consider $\beta(s) = \beta_0$, i.e., a constant temperature, for which we have $\gamma(t) = \Theta(\exp(-\beta_0\exp(-\beta_0)t))$. Provided that $\beta_0$ is not too large or too small, the modality gap closes quickly at a linear rate. More generally, a linear schedule $\beta(s) = (1 - (s/t))\beta_0 + (s/t)\beta_1$ also yields a linear decay $\gamma(t) = \exp(-c(\beta_0, \beta_1)t)$ where $c(\beta_0, \beta_1) = [(\beta_0 + 1)\exp(-\beta_0) - (\beta_1 + 1)\exp(-\beta_1)]/(\beta_1 - \beta_0)$, while allowing a sweep of temperature values throughout training.

# D. Methods

In this section, we introduce the proposed methods in greater details, and present representative results in Table 2 as examples. We include more ablation studies in Appendix F.

## D.1. Temperature Control

**Temperature Scheduling (TS).** First, we propose to not learn the temperature as [1]. Instead, we schedule the increase of temperature according to the training epochs *linearly*. With a larger temperature in the late stage, we can make more progress on reducing the modality gap. As shown in Table 2, linearly increasing temperature from $10^{-2}$ to $5 \cdot 10^{-2}$ over the training process reduces the modality gap and leads to better text-image retrieval performance.

**Temperature Reparameterization (TR).** Towards the same purpose, TR designs different parameterization $\tau(\nu)$ from CLIP to slow down the decreasing of $\tau(\nu)$ when learning $\nu$. In comparison with the original CLIP with $1/\tau(\nu) = \exp(\nu)$, we propose to use $1/\tau(\nu) = \exp(\nu/s)$ with $s > 1$ being a scalar to reduce the influence of $\nu$ as it grows. Another way we propose is to design through softplus to achieve the same goal: $1/\tau(\nu) = \log(1 + e^\nu)$. We report results using softplus in Table 2.

**Smaller Learning Rate of Temperature (SLRT).** To slow down the decrease of $\tau(\nu)$, SLRT scales down the learning rate of $\nu$ directly. In Table 2, we report the results from scaling the learning rate of $\nu$ by $10^{-1}$. The result shows improved performance with reduced gap. It implies that the original choice of $\nu$ in CLIP is far from optimal, and can be improved through a careful analysis.

**Fixed on Large Temperature (FLT).** FLT freezes the temperature instead of learning it. We let $\tau(\nu)$ range from $10^{-2}$ to $4 \cdot 10^{-1}$ to find temperatures that can shrink the modality gap. The results from Table 2 is obtained with $\tau(\nu) = 4 \cdot 10^{-2}$.

## D.2. Swap Modalities

**Hard Swapping Between Modalities (HS).** Recall $\boldsymbol{H}_X, \boldsymbol{H}_Y \in \mathbb{R}^{n \times d}$ denote the image and text features. HS randomly swaps $\boldsymbol{H}_X[i, j]$ and $\boldsymbol{H}_Y[i, j]$ for each $(i, j)$ independently with probability $0.5$ to obtain $\overline{\boldsymbol{H}}_X, \overline{\boldsymbol{H}}_Y$. *Then, the swapped features are input to the loss function for optimizing networks.* Such swapping only happens for $p$ random portion of the training, thus $p$ controls how much swapping is applied across training. We have conducted search on $p$ from $1e - 3$ to $1.0$. It turns out to be a strong method for closing the gap. The results from Table 2 are obtained with $p = 1e - 3$.

**Soft Swapping Between Modalities (SS).** SS is similar to HS, but swaps entries in a soft way. In SS, we randomly sample $\lambda_{ij} \in [0, 1]$ for each $(i, j)$ independently, and then set $\overline{\boldsymbol{H}}_X[i, j] = \lambda_{ij}\boldsymbol{H}_X[i, j] + (1 - \lambda_{ij})\boldsymbol{H}_Y[i, j]$ and $\overline{\boldsymbol{H}}_Y[i, j] = \lambda_{ij}\boldsymbol{H}_Y[i, j] + (1 - \lambda_{ij})\boldsymbol{H}_X[i, j]$. Similarly, the swapped features are input to the loss function for optimizing networks. Besides, swapping is also conducted for $p$ random portion of the training. We have conducted a search on $p$ from $1e - 2$ to $1e - 1$ The result from Table 2 is obtained with $p = 5e - 2$.

| | Baseline | Controlling the Temperature | | | | Eliminating the Separation of Modalities | |
|---|---|---|---|---|---|---|---|
| | | FLT | TS | TR | SLRT | HS | SS |
| Modality Gap ($\downarrow$) | 0.294 | 0.122 | **0.088** | 0.284 | 0.268 | 0.225 | 0.139 |
| Uniformity ($\uparrow$) | -1.165 | -1.183 | **-1.108** | -1.153 | -1.152 | -1.156 | -1.152 |
| Zero-shot ($\uparrow$) | 13.650 | 14.400 | **17.167** | 14.860 | 15.130 | 13.910 | 14.330 |
| Linear Probe ($\uparrow$) | 65.550 | 61.980 | **67.500** | 66.460 | 66.630 | 65.950 | 64.940 |
| Image Retrieval ($\uparrow$) | 68.233 | 73.440 | **75.723** | 70.060 | 70.320 | 69.716 | 68.912 |
| Text Retrieval ($\uparrow$) | 78.367 | 82.660 | **85.313** | 80.100 | 80.400 | 79.740 | 79.160 |

Table 2: **Mitigating the modality gap with our proposed methods.**

# E. Experiment Settings

## E.1. Pretraining

**Basics** For the CLIP pretraining on MSCOCO using the train split, we utilize the OpenCLIP plat-form produced by [50]. The neural network architecture for the image encoder is ResNet-50 [51] and the architecture for the text encoder is Transformer [52]. Both architectures can be specified by 'RN50' in the OpenCLIP platform. The input image is RandomResizedCrop to $224 \times 224$ followed by Normalization. For each method of pretraining, we select the best learning rate from $[1e-5, 5e-5, 1e-4, 5e-4]$. For each training process, we train the model for $100$ epochs with batchsize $1024$ using a cosine decay learning rate schedule and 20 warmup epochs. We conduct the training 3 times for each specific setting and report the mean of the results among all 3 trainings.

**Computing Requirements** All the experiments can be conducted on a single A100 GPU.

## E.2. Metrics

**Modality Gap** We measure the modality gap as modality center distance. The center distance is calculated between $512$ random pairs in the MSCOCO test split. Let $\boldsymbol{X}$ and $\boldsymbol{Y}$ be image and text features, where each column is a feature sample, we define the modality centers as

$$\boldsymbol{c_X} = \frac{1}{n} \sum_{j=1}^{n} \boldsymbol{x}_j, \boldsymbol{c_Y} = \frac{1}{n} \sum_{j=1}^{n} \boldsymbol{x}_j.$$

Then as proposed in [12], we measure modality gap by the center distance:

$$m_{\boldsymbol{XY}} = \|\boldsymbol{\Delta}\| = \|\boldsymbol{c_X} - \boldsymbol{c_Y}\|.$$

The center distance reported is also averaged among three independent trainings. A larger center distance indicates a larger modality gap.

**Uniformity** We utilize the uniformity metrix proposed by [31]. After concatenating $\boldsymbol{X}$ and $\boldsymbol{Y}$ and obtain $\boldsymbol{Z} \in \mathbb{R}^{d \times 2n}$, we define the sample mean of $\boldsymbol{Z}$ as $\widehat{\boldsymbol{\mu}}$ and the sample variance of $\boldsymbol{Z}$ as $\widehat{\boldsymbol{\mu}}$. Calculate the quadratic Wasserstein distance between $\mathcal{N}(\widehat{\boldsymbol{\mu}}, \widehat{\boldsymbol{\Sigma}})$ and $\mathcal{N}(\boldsymbol{0}, \mathbf{I}/m)$:

$$\mathcal{W}_2 := \sqrt{\|\widehat{\boldsymbol{\mu}}\|_2^2 + 1 + \text{tr}(\widehat{\boldsymbol{\Sigma}}) - \frac{2}{\sqrt{m}} \text{tr}\left(\widehat{\boldsymbol{\Sigma}}^{\frac{1}{2}}\right)}.$$

Then $-\mathcal{W}_2$ serves as the uniformity metric. A larger value of $-\mathcal{W}_2$ indicates a better uniformity of the feature space spanned by $\boldsymbol{Z}$.

## E.3. Downstream Tasks

**Linear Probe** We conduct linear probing on CIFAR 10 following the same setting in [1]. After pertaining, we extract the image features from the vision encoder, and train a logistic regression

| | Uniformity | Modality Gap | Zero Shot | Linear Probe | Image Retrieval | Text Retrieval |
|---|---|---|---|---|---|---|
| Baseline | -1.165 | 0.294 | 13.65 | 65.55 | 68.23 | 78.36 |
| 1.00E-02 | -1.214 | 0.669 | 18.48 | 69.44 | 69.89 | 80.68 |
| 2.00E-02 | -1.138 | 0.334 | 18.83 | 68.41 | 70.54 | 81.3 |
| 4.00E-02 | -1.183 | 0.122 | 14.4 | 61.98 | 73.44 | 82.66 |
| 7.00E-02 | -1.246 | 0.033 | 11.76 | 49.61 | 75.53 | 82.88 |
| 1.00E-01 | -1.251 | 0.019 | 10.42 | 48.29 | 66.55 | 73.58 |
| 2.00E-01 | -1.255 | 0.008 | 10.55 | 50.14 | 23.97 | 26.94 |
| 3.00E-01 | -1.282 | 0.007 | 10.69 | 42.09 | 10.4 | 12.8 |
| 4.00E-01 | -1.296 | 0.009 | 13.83 | 37.97 | 6.9 | 8.7 |

Table 3: **More results for fixed temperature (FLT).**

| | Uniformity | Modality Gap | Zero Shot | Linear Probe | Image Retrieval | Text Retrieval |
|---|---|---|---|---|---|---|
| Baseline | -1.165 | 0.294 | 13.65 | 65.55 | 68.23 | 78.36 |
| S0 | -1.120 | 0.384 | 18.40 | 69.40 | 72.55 | 82.89 |
| S1 | -1.087 | 0.198 | 18.55 | 69.26 | 72.69 | 83.05 |
| S2 | -1.092 | 0.137 | 16.80 | 68.25 | 73.98 | 84.25 |
| S3 | -1.108 | 0.088 | 17.17 | 67.50 | 75.72 | 85.31 |
| A0 | -1.151 | 0.541 | 19.50 | 69.47 | 72.51 | 81.97 |
| A1 | -1.120 | 0.261 | 16.85 | 67.46 | 73.90 | 83.65 |

Table 4: **More results for temperature scheduling (TS).**

based using the features in the train split. We evaluate the classification accuracy of the features in the test split and report the averaged results across three independent trainings. We report the Top-1 classification accuracy.

**Zero-shot Classification** We conduct linear probing on CIFAR 10 following the same setting in [1]. For each class containing [NUMBER], we construct the text prompt as "A photo of the number [NUMBER]". For each image, the text encoder will embed the text prompts into text features for all classes, and the image encoder will embed the image into an image feature. The probability of each class is then obtained from the softmax of the cosine similarities between the text features and the image feature scaled by temperature. We report the Top-1 classification accuracy. All values are averaged across three trainings.

**Text-image Retrieval** We conduct image-to-text retrieval and text-to-image retrieval on the text split of MSCOCO, and report the Top-1 recall. All values are averaged across three trainings. Image-to-text retrieval aims to retrieve the most relevant text description given an input image by computing the similarities of image and text features, while text-to-image retrieval is performed the other way around.

**MMVP-VLM** The MMVP-VLM benchmark is proposed by [22], which can be viewed as a "difficult retrieval task". Given two image and text pairs $(x_1, y_1)$ and $(x_2, y_2)$ that only have differences concerning certain detailed visual pattern, the CLIP model is required to perfectly match the two pairs. They define 9 visual patterns: Orientation and Direction, Presence of Specific Features, State and Condition, Quantity and Count, Positional and Relational Context, Color and Appearance, Structural and Physical Characteristics, Text, and Viewpoint and Perspective. Each visual pattern contains several text-image pairs for constructing the matching task. We report the average accuracy across 9 visual pattern groups. All results are averaged across three trainings.

# F. Additional Experimental Results

In this section, we provide more detailed results on different methods that reduce (or enlarge) the modality gap. We introduce details of the results below.

|  | Uniformity | Modality Gap | Zero Shot | Linear Probe | Image Retrieval | Text Retrieval |
|---|---|---|---|---|---|---|
| Baseline | -1.165 | 0.294 | 13.65 | 65.55 | 68.23 | 78.36 |
| Softplus | -1.153 | 0.284 | 14.86 | 66.46 | 70.06 | 80.1 |
| Scale2 | -1.154 | 0.287 | 18.25 | 66.3 | 70.32 | 80.6 |
| Scale10 | -1.153 | 0.285 | 14.36 | 66.5 | 70.15 | 80.3 |
| HS P1 | -1.317 | 0.028 | 14.36 | 39.72 | 47.17 | 53.22 |
| HS P1e-1 | -1.127 | 0.046 | 17.39 | 56.88 | 67.1 | 75.8 |
| HS P1e-2 | -1.181 | 0.095 | 15.7 | 64.29 | 69.36 | 79.4 |
| Hard Swap Feature | -1.155 | 0.289 | 14.71 | 65.91 | 69.93 | 80.6 |
| Remove Normalization | -1.005 | 0.468 | 16.82 | 65.4 | 58.62 | 74.16 |

Table 5: **More results for other methods.**

- More results for fixed temperature: we provide a more fine-grid temperature search from $1e^{-2}$ to $4e^{-1}$, and present the results in Table 3.

- More results for temperature scheduling: The three temperature schedules S1, S2, S3 presented in Table 1 in Section 4.1 have the following schedules. S1 corresponds to linearly increasing temperature from $1e-2$ to $3e-2$, S2 corresponds to corresponds to linearly increasing temperature from $1e-2$ to $4e-2$, and S3 corresponds to linearly increasing temperature from $1e-2$ to $5e-2$. Apart from the above scheduling, we additionally show results for S1, A0, and A1 in Table 4. Here, S0 corresponds to linearly increasing temperature from $1e-2$ to $2e-2$, A1 corresponds to the alternation of temperature between $1e-2$ and $2e-2$, and A2 corresponds to the alternation of temperature between $1e-2$ and $4e-2$. Temperature alternation follows a cosine function.

- More results for other methods: More results for other methods are shown in Table 5. Here, Scale2 and Scale10 corresponding to temperature parameterization with $\tau(\nu) = \exp(\frac{\nu}{s})$, where $s = 2, 10$, respectively. HS P* refers to Hard Swapping with $p = *$ as discussed in Appendix D. Besides, Hard Swap Feature means swapping the entire row instead of independent entries. It can also break the modality boundary but with greater constraints. As we can see, Hard Swap Feature reduces the center distance as well, but the reduced amount is less than HS (swapping entries). Lastly, Remove Normalization means freezing the temperature while removing the normalization of features during training. This approach is equivalent to learning a "temperature parameter" for each feature, in the sense that the temperature parameter is a global scaling of feature norms. With more freedom in changing the "temperatures" to optimize losses, Remove Normalization results in a larger modality gap.

