# OpenReview forum: "Explaining and Mitigating the Modality Gap in Contrastive Multimodal Learning"
_CPAL.cc/2025/Proceedings_Track — CPAL 2025 (Proceedings Track) Poster_

### Official Review · Reviewer_bXj8 · 2025-01-07

**Rating:** 8
**Confidence:** 4

**Review:**

This paper explores the modality gap in contrastive multimodal learning. In particular, this paper explains the reason why the modality gap exists in the training process of CLIP and further provides basic solutions to mitigate the modality gap. Most importantly, this paper provides theoretical analysis and practical experiment verification. Both the theoretical and practical contributions deliver a better understanding of the modality gap that naturally exists in multimodal learning.

Pros:
1. The paper shows a unique view to dive into the modality gap in multimodal learning. This paper considers the gradient dynamics in the CLIP training process and provides one view to understand the modality gap caused by the mismatched image-text pair. Both the theoretical analysis and practical experiments deliver a better understanding of the modality gap.
2. Both the theoretical and practical contributions present a potential metric to evaluate the modality gap and some insights to mitigate the gap.
3. The comprehensive experiments show mitigating the modality gap does not always work well for every multimodal learning task, which means this paper mainly provides a unique view to understand the modality gap and how to mitigate the gap based on the gradient flow dynamics.
4. The conclusion is fair and reasonable, which is also aligned with some previous and latest work. There are some tasks where mitigating the gap is better [1], while some tasks strengthen the dominant modality [2]. This paper gives another unique view to understand the reason.
[1] “The modality focusing hypothesis: Towards understanding crossmodal knowledge distillation”, ICLR 2023
[2] “DLF: Disentangled-Language-Focused Multimodal Sentiment Analysis”, AAAI 2025

Cons:
1. This paper considers the CLIP model, which contains two modalities, image and text. If the models or tasks contain 3 modalities, whether the theoretical analysis and conclusion cover those applications.

Questions:
1. There are many factors in multimodal learning, such as alignment and fusion. Why this paper dives into the gradient flow dynamics and the temperature parameter. What is the motivation behind this choice?

---

### Official Review · Reviewer_zLXi · 2025-01-13

**Rating:** 6
**Confidence:** 4

**Review:**

Summary:

The manuscript investigates the emergence of the modality gap in multimodal learning models, such as CLIP, which map images and text into a shared feature space. The study analyzes the gradient flow dynamics during training and identifies mismatched data pairs and a learnable temperature parameter as key contributors to the modality gap. The results demonstrate that mitigating the modality gap through strategies like 1) temperature scheduling and 2) modality swapping improves performance on tasks like image-text retrieval. The proposed work also provides theoretical insights for enhancing multimodal learning frameworks.


Strengths:
The paper provides analysis of the modality gap in multimodal models like CLIP, demonstrating that the gap diminishes at an extremely slow rate during training and is influenced by learned temperature parameters. The authors also explain why the modality gap is inherent at initialization and cannot be fully closed before training to offer valuable theoretical guidance for addressing this issue.

Building on their theoretical findings, the authors propose practical solutions, including temperature scheduling and feature exchange between modalities to effectively reduce the modality gap. These methods are validated across multiple downstream tasks, showcasing their utility in improving image-text retrieval performance.

Weaknesses:

In Section 4.1, the method mentions about the the methods to control the temperature during training. However, it is not clear to me and I wonder if the authors could give more intuition about the reasons why the temperature controlling strategies work for the tasks in the paper. Clarification is needed these strategies.

---

### Meta-Review · Area_Chair_dx9F · 2025-02-05

**Recommendation:** Accept (Poster)
**Confidence:** 4

**Metareview:**

This paper explores the modality gap in contrastive multimodal learning by (1) explaining the reason why the modality gap exists in the training process of CLIP and (2) further providing basic solutions to mitigate the modality gap. Both theoretical analysis and practical experiment verification are provided. All reviewers acknowledge the significance of this research.

---

### Decision · Program_Chairs · 2025-02-11

Accept (Poster)